# Geographic Information Systems (GIS) to Assess Dental Caries, Overweight and Obesity in Schoolchildren in the City of Alfenas, Brazil

**DOI:** 10.3390/ijerph20032443

**Published:** 2023-01-30

**Authors:** Iago Ramirez, Diego Escobar Alves, Patrick Calvano Kuchler, Isabela Ribeiro Madalena, Daniela Coelho de Lima, Mariane Carolina Faria Barbosa, Maria Angelica Hueb de Menezes Oliveira, Geraldo Thedei Júnior, Flares Baratto-Filho, Erika Calvano Küchler, Daniela Silva Barroso de Oliveira

**Affiliations:** 1Department of Clinic and Surgery, School of Dentistry, Federal University of Alfenas, Alfenas 37130-001, Brazil; 2Institute of Geography, Department of Physical Geography, Rio de Janeiro State University (UERJ/IGEOG/DGF), Rua São Francisco Xavier, 524, Maracanã, Rio de Janeiro 25550-013, Brazil; 3Department of Biomaterials, University of Uberaba, Uberaba 35430-026, Brazil; 4Department of Dentistry, University of Joinville Region, Joinville 89219-710, Brazil; 5School of Dentistry, Presidente Tancredo de Almeida Neves University Center, São João del Rei 66645-057, Brazil; 6School of Dentistry, Tuiuti University from Parana, Curitiba 82010-210, Brazil

**Keywords:** child, dental caries, nutrition

## Abstract

Childhood-related obesity and overweight are increasing concerns for the health and well-being of children. Dental caries (decay) is the most prevalent oral disease during childhood, and several studies have suggested that nutritional status and dental caries are associated in children. Therefore, this study aimed to determine the geographic distribution of childhood overweight/obesity and dental caries in a medium-sized Brazilian city. This cross-sectional study was conducted with 269 children of both genders enrolled in four public schools in the city of Alfenas. The children were clinically examined to assess cavitated dental caries and nutritional status (overweight and obesity). In addition, the GIS was used for the geospatial clustering analyses. A heat map was created by the Kemel method to estimate the concentration of the outcomes. The cavitated dental caries and overweight/obesity were also pointed out by dots on the map. However, of the 269 children, 118 were boys (43.87%) and 151 were girls (56.13%). One hundred fifty-seven children (58.4%) were classified as having “non-cavitated caries,” while 112 (41.6%) were classified as having “cavitied caries.” In the nutritional status assessment, 204 children (75.84%) were classified as “eutrophic,” while 65 children (24.16%) were classified as “overweight/obesity,” A geographical correlation of dental caries with overweight/obesity may exist in the northeast and southwest areas. In conclusion, a geographical concordance between the dental caries and the occurrence of overweight/obesity among the schoolchildren from Alfenas may exist in some areas. Future studies are necessary.

## 1. Introduction

The most prevalent and consequential oral diseases in childhood globally are dental caries, which affect the quality of life and social interactions and are directly associated with pain, dysfunction, and aesthetic problems [1]. A systematic review and meta-analysis published in 2020 explored the prevalence of dental caries in children worldwide and investigated studies conducted from 1995 to 2019. The meta-analysis included a total of 164 articles and observed that the prevalence of dental caries in deciduous teeth in the world in a sample size of 80,405 was 46.2% and the prevalence of dental caries in permanent dentition in children in the world in a sample size of 1,454,871 was 53.8% [2]. This global prevalence and distribution of dental caries reflect the social inequalities related to its predisposing and etiological factors [3]. These numbers show that although dental caries decreased in the past century, it continues to be a public health problem worldwide. On the other hand, in the past years, childhood obesity and also overweight have increased expressively in many geographic areas, and consequently became a major public health issue recently. Furthermore, childhood-related obesity and overweight are an increasing concern with regard to the health and well-being of the child [4] and the association of these conditions with adverse health consequences throughout the life-course [5].

Dental caries has been extensively explored and associated with childhood obesity in the literature. Several systematic reviews have pulled studies from different populations that investigated the association between body mass index and dental caries [1,6,7]. Although much research investigating the association between these two conditions has been conducted for many years, primary studies and systematic reviews show divergent results once obesity and/or overweight have been associated with higher [6,8,9] and lower risk for dental caries [6,10,11,12,13].

In our previous study in Brazilian children, we observed that overweight and obese children present less untreated dental caries are than normal weight children. The fact that untreated dental caries are lower in overweight/obesity children could be explained by socioeconomic and demographic factors [13]. Thus, to explore the interplay between the demographic characteristics of these two conditions (dental caries and a high body mass index) is necessary for intervention strategies in public health. In this context, the Geographic Information System (GIS) is a helpful tool to evaluate the geographic distribution of health conditions [14], allowing the examination of local geographic variations and the identification of hot spots. The GIS contributes also to quantifying the spatial distribution of a given condition, such as dental caries [15] and obesity [16], and to investigating differences in the spatial correlation between dental caries and obesity. Therefore, this study aimed to determine the geographic variation of childhood overweight/obesity and dental caries in the city of Alfenas, Minas Gerais State, southeast Brazil. It also aimed to identify regions with significantly high and low clusters of these conditions and explore if their association has a relationship with demographic characteristics, identifying the concordance and discordance in the geographic distribution of untreated dental caries and body mass index variation.

## 2. Materials and Methods

This is a cross-sectional study, that is part of a larger project that evaluates oral and dental traits in schoolchildren from Alfenas (Minas Gerais), southeast Brazil. The sample size was determined based on the total number of children (age 8–11 years) enrolled in the public schools in the urban area of Alfenas City. The calculation used a power of 80% and an alpha of 0.05 and predicted a minimum of 276 children. The total number of included children was equally divided among the four urban schools [13]. This study was approved by the Human Ethics Committee of the Federal University of Alfenas (78568217.7.0000.5142). The present study was conducted in accordance with the Declaration of Helsinki. The legal guardians agreed to the child’s data inclusion in the study with written informed consent. However, an assent document was used for children.

This study was conducted to georeference the sample studied by Barbosa et al. [13] in order to investigate the spatial distribution profile of dental caries and overweight/obesity and also to explore the association between both conditions. Therefore, the data from the schoolchildren enrolled in our previous study [13] were screened, and all children that had their residence addresses correctly referred to and officially registered within the City Hall for the map confection were included in this study.

The sample consisted of children enrolled in 4 public schools in the city of Alfenas, distributed throughout 8 districts. Alfenas is a city that is located in the southeast region of Brazil, in Minas Gerais state. It is classified as a medium-sized city with a population of 73,774 in 2010, which was the last census in Brazil [17].

Some characteristics of these children were previously described [12,13,18,19]. The children’s oral examination and nutritional status evaluation, as well as the data collection, were done in 2018 by one single examiner, trained and calibrated, and an assistant writer assistant. The Kappa intra-examiner for dental caries was very good (Kappa = 0.87).

The examination of untreated dental caries was done inside the school, using cotton rollers, gauze, standard mouth mirrors, and a ball-point probe under natural light (World Health Organization—WHO) [20]. 

The dental caries lesions were diagnosed by visual examination according to the ICDAS (International System for Detection and Assessment of Carious Lesions) criteria that uses an ordinal scale (0 to 6), ranging from the healthy surface to extensive cavitation [21]. Dental caries was determined as “non-cavitated dental caries” or “cavitated dental caries,” in which “non-cavitated dental caries” refers to the 0–2 score and “cavitated dental caries” refers to the grouping of 3–6 scores [21]. 

The nutritional status was calculated by the Body Mass Index (BMI), in which the height and weight were recorded on the day of the dental caries examination. The height was evaluated in meters with the children erect against a wall-mounted scale. The weight was evaluated in kilograms with children wearing no shoes and light clothes, with precision varying from to nearest 0.01 kg. The BMI was calculated for each child using the z-score calculator (http://zscore.research.chop.edu/index.php (accessed on 3 May 2018), considering age and gender, and obtaining the classifications of nutritional status as recommended by the WHO [22]. In order to categorize the nutritional status in this study, the children were classified as “eutrophic” (BMI z-score ≥ 3 and ≤ 85th percentile), which is the normal weight group, or as “overweight/obesity” (BMI z-score > 85th percentile). The classification of overweighting and obesity was based on the fact that these two nutritional statuses presented the same trend in our previous evaluation reported in Barbosa et al. [13].

Children classified as “underweight” (BMI z-score < 3 percentile) were excluded from the study.

A chi-square was used to compare dental caries experience among the nutritional status groups using GraphPad Prism (GraphPad Prism 5.0, San Diego, CA, USA). The established alpha was 5%.

### GIS and Maps Creations

The sample for the geospatial clustering analyses was selected from a previous database [13]. The data for dental caries and BMI classification was the basis for the construction of a GIS and subsequent representation of the events in maps, which allowed the spatial analysis. The software QGIS (V. 3.26.2, Open-Source Geospatial Foundation Project, 2021), an open-source multi-institution initiative (QGIS, 2022), was used for the GIS. All children included in the sample had their residence addresses correctly referred and officially registered at City Hall for the maps. Through the attributes related to the address of each child, the geoprocessing of the tables was done. The Open Street Map (OSM) database was used to process the addresses by geocoding. The addresses obtained were matched with the OSM database in order to provide an adequate use for the addresses.

Kernel estimation (heat map) was used to estimate the concentration of the overweight/obesity events that occurred at different places in order to analyze spatial clustering and spatial patterns. For the kernel estimation, it is important to define its operating radius, since this is where the impact area of the heat map will be reflected. Additionally, for the purpose of definition, some tests were produced to better choose the region of representation of the data, with a chosen radius of 200 m. Subsequently, the values reached were classified by the Natural Break method (Jenks), according to Ribeiro e Silva [23]. The outcome of dental caries was marked on the maps by dots, marking the occurrence of “Yes” or “No” in different colors. 

The main outcome was the identification of a concordance or discordance relation between the nutritional status (“Eutrophic” or “Overweight/Obesity”), and dental caries (“yes”—cavitated dental caries or “no”—non-cavitated dental caries). This was identified through the analysis of hotspots or dot concentrations in a geospatial cluster that overlapped.

## 3. Results

Among the 353 screened children’s data, 15 children were classified as “underweight,” and 46 children who did not present a complete address or for whom it was not possible to map the address were excluded. Additionally, 23 people who live in the rural area of Alfenas were not included in the present analysis. Therefore, the final sample included 269 children of both genders (age ranging between 8 and 11 years), of whom 118 were boys (43.87%) and 151 were girls (56.13%). One hundred fifty-seven children (58.4%) were classified as having “non-cavitated caries,” while 112 (41.6%) were classified as having “cavitied caries” for the presence of dental caries cavitated lesions. Table 1 shows the distribution of the characteristics according to the groups. In the nutritional status assessment, 204 children (75.84%) were classified as “eutrophic,” while 65 children (24.16%) were classified as “overweight/obesity,” 

Table 2 shows the statistical association between dental caries and nutritional status. Only a statistically borderline association between dental caries and overweight/obesity was not observed (*p* = 0.0798).

The Figure 1 presents the overweight/obesity and eutrophic, and the cavitated dental caries and non-cavitated dental caries cases’ dispersion. This map was created with the purpose of facilitating understanding and comparison (contextual and individual). Cavitated dental caries and overweight/obesity outcomes were observed in a dispersive way. The spatial distribution suggested that in the northeast and southwest areas, cavitated dental caries and overweight/obesity were more closely correlated geographically, while in the central area, only overweight/obesity was highly dense. Figure 1 shows the georeferencing of the space distribution of the residences of the children according to the outcomes. 

The locations of the public dental care services were not spatially correlated with the occurrence of untreated cavitated dental caries, suggesting that regional accessibility to dental treatment was not a factor involved in untreated dental caries. 

## 4. Discussion

Dental caries and childhood overweight/obesity epidemics are multifactorial complex diseases, in which children’s dietary pattern is reported as a common underlying etiologic factor. As a result, in populations where obese children present more dental caries, a logical connection for both conditions is the high ingestion of added sugar intake, which is a common factor for dental caries [9,24,25,26] and overweight/obesity [8,24,26,27,28,29]. However, some studies observed an inverse association, in which obese children have lower or inconclusive dental caries experiences [6,10,11,30], as we observed in our previous study investigating this population of schoolchildren from Alfenas. In our evaluation report in Barbosa et al. [13], we observed that untreated dental caries was more common in eutrophic children than in overweight/obese children, with statistical significance. This finding is in agreement with the results observed in 801 schoolchildren from the south of Brazil, which also observed an inverse association between caries and obesity [31]. Therefore, our hypothesis for this “inversed association” observed in Brazilian schoolchildren lies in the socio-economical connection, in which families with higher income might have more access to sweets and candies, oral health products, and also dental treatment. In this scenario, our hypothesis is that clusters of obesity and lower dental caries exist and would be observed in areas with higher incomes. 

In order to explore the spatial distribution and possible overlap of overweight/obesity and caries experiences, we reexamined the data from schoolchildren in Alfenas. Alfenas is classified as a medium-sized county located in Brazil with a population of 79,996 inhabitants, a controlled adjustment of fluoride to a public water supply [32], a score of 0.761 HDI, and an 8.08/1000 child mortality rate [17]. Our spatial analysis showed a geographical variation of overweight/obese children in different regions, indicating the potential presence of spatial clusters or spatial heterogeneity. The analysis of the maps of the urban region of Alfenas suggested that the hot spots of overweight/obesity correlated with cavitated dental caries and occurred distant from the central area (the northeast and southwest areas). This data matches with other cross-sectional epidemiological studies that have reported the most socioeconomically vulnerable population located at the neighborhoods [33,34,35]. The populations distant from central areas are commonly associated with a vulnerable economic status, a higher risk for environmental-associated diseases, and inadequate nutrition [36]. 

Additionally, our initial hypothesis was that a spatial analysis would reveal overlapping hotspots of children with non-cavitated dental caries and overweight/obese. However, the interpretation of this data is more complex, suggesting that the central area presents a different pattern from the northeast and southwest areas. In the central area, a high density of overweight/obesity and a low density of cavitated dental caries were noted. This suggested that in the higher income area, a low caries experience is observed overlaps with overweight/obesity. It is important to emphasize that dental caries [36] and overweight/obesity [1] are intimately related to life habits, which could explain the different results observed according to the geographic area and in the studies that explored the association between nutritional status and dental caries. This makes the approach to studying these chronic diseases even more complex. Furthermore, the adherence of protective dietary habits is a difficult challenge to achieve in schoolchildren [37].

Several studies have been demonstrating that dental caries is still highly incident and prevalent in many regions of Brazil [38,39,40,41,42], therefore, it is necessary to identify the population and the risk factors involved with a higher occurrence risk in order to trace an efficient surveillance. Boing et al. [36] have identified that dental caries has higher rates among the poorest individuals, and the identification of populations at higher risk may support interventional strategies to minimize the incidence of this condition [43]. Our results also suggested that regional accessibility to dental treatment was not a factor involved in untreated dental caries. It is important to clarify that the dental care services shown here are public and free. 

The limitations of our study should be raised here. The fact that the sample included only schoolchildren from the public educational system from schools located only in urban areas is a limitation, since children enrolled in private schools are not included. In addition, we evaluated only children 8 to 11 years old, ideally, children of different ages should be investigated. Another limitation of our study is the absence of other information, such as dietary factors or income factors. Future studies should be performed using a larger population of children of different ages and also using the future Brazilian census data. The last Brazilian census occurred in 2010, and the results from the next Brazilian census, which is expected for this year, will allow a better understanding of the factors involved in dental caries and nutritional status in different geographic areas. Historically, dental caries tends to be distributed with greater concentration in a small portion of the population [36]. This complex relationship between dental caries and its contextual factors composes a paradigm for oral health epidemiology [44]. 

Furthermore, even with better oral health status in children over the past decade [45], the association between dental caries and a healthy diet [37] and environmental/geographic factors still makes it a major challenge for the public’s health [3]. In addition, the prevalence of overweight and obesity, especially among children, must be observed as a consistent signal to the health authorities to intervene with health education programs [5]. However, with early intervention, a variety of chronic conditions and comorbidities can be avoided in adulthood [1]. These two conditions, as well as the factors involved in their association, should be widely investigated. 

## 5. Conclusions

In this study, the visualization of the map showed a dispersion distribution of the studied outcomes, especially overweight/obesity. It is possible that a geographical concordance of the occurrence of overweight/obesity and cavitated dental caries among children from public schools in Alfenas (Brazil) exists in specific regions; however, future studies using larger data sets are necessary to confirm this trend. 

## Figures and Tables

**Figure 1 ijerph-20-02443-f001:**
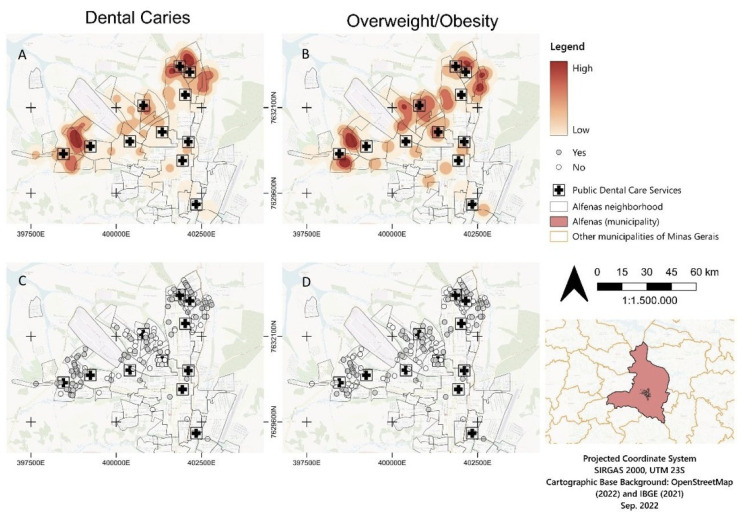
Maps representing the comparison of georeferencing of space distribution of residences of the children. (**A**) Cavitated dental caries represented by the heat map. (**B**) Overweight/obesity represented by the heat map. (**C**) Cavitated dental caries represented by the dots (each child is represented by a dot). (**D**) Overweight/obesity represented by the dots (each child is represented by a dot). The public dental care services are represented by the squares with the cross inside.

**Table 1 ijerph-20-02443-t001:** Distribution of the characteristics according to the groups.

Variable	Dental Caries		Nutritional Status	
	Non-Cavitated Caries	Cavitated Caries	Eutrophic	Overweight/Obesity
Total of sample n (%)	157 (58.4)	112 (41.6)	204 (75.8)	65 (24.2)
Gender				
Male n (%)	68 (25.3)	50 (18.6)	87 (32.3)	31 (11.5)
Female n (%)	89 (33.1)	62 (23.0)	117 (43.5)	34 (12.6)
Mean age (SD *)	8.85 (0.79)	8.77 (0.71)	8.80 (0.88)	8.81 (0.81)

Note: * SD means standard deviation.

**Table 2 ijerph-20-02443-t002:** Overweight/obesity distribution according to groups.

GROUPS	Non-Cavitated Caries n (%)	Navitated Caries n (%)	*p*-Value
Eutrophic	113 (42.0)	91 (33.8)	0.0798
Overweight/obesity	44 (16.4)	21 (7.8)	

Note: Chi-square was used.

## Data Availability

Data available upon request.

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
