# Peer review of "Geographic Information Systems (GIS) to Assess Dental Caries, Overweight and Obesity in Schoolchildren in the City of Alfenas, Brazil"

_ijerph, 2023, doi:10.3390/ijerph20032443_

Round 1

Reviewer 1 Report

Introduction: Comprehensive. It prepared the reader as to why the study was done and what the authors wanted to investigate.  

Methods: A unique open-source correlation tool (Geographic Information Systems or GIS) was used to ‘co-map’ caries prevalence (measured by school survey using natural light and calibrated examiners to record the presence or absence of caries using ICDAS) and obesity (as measured by BMI). School dental inspections are standard for public health caries surveys but not comprehensive (i.e. caries diagnosis was not carried out by a dentist, using intra-oral imaging, caries detection devices for occlusal, smooth surface and interproximal caries, such as DIAGNOdent, Carivu or digital radiography). The authors claim that their examiners were able to detect all scores of ICDAS but that is difficult to imagine unless they truly did have an intense light (dental unit light or head lamp) and compressed air to dry the teeth for 5 seconds as claimed.  Categories 1, 2, and 3 ICDAS are very hard to see by just looking in the mouth. See (https://pubmed.ncbi.nlm.nih.gov/17518963/). More details about the examination process and calibration of the examiners would be helpful.

Statistical analysis: There was no attempt at formal statistical testing of the (non)correlation claimed between caries prevalence and obesity. This was judged simply by looking at the maps that were produced. It would have been nice to have at least some kind of statistical confirmation that

1.     dental caries outcomes were observed in a dispersive way, with no significant concentration at the downtown or in the neighborhoods.

2.     The location of the public dental care services was not spatially correlated with the occurrence of untreated dental caries

Results: Results are based on just viewing the correlation maps. While this may be convincing to the trained research, to convince the reader, simple statistical analysis would be helpful. I would encourage the authors to attempt some kind of basic statistical analysis. See Yen et al (reference [34]) for a guide as to which statistic tool to use.

Discussion and Conclusion: This reviewer is convinced by the map shown in Figure 1 clearly showing no correlation between caries and obesity but this needs to be confirmed with statistical analysis. More discussion as to why these results were obtained would help as well. The pattern of sugar consumption may be paramount, rather than the total daily intake of excess calories contributing to obesity. Are there any habits of sugar consumption that could provide some clues as to why eutrophic children have just as many dental caries as obese children? Can that even be stated without the statistical analysis?

Reviewer recommendation: Revise and resubmit.

Reviewer 2 Report

Thanks for submitting this manuscript.

Some of the comments need to be addressed in this manuscript mentioned below:

1. The topic of the study seems to be interesting, but the title is very elaborative, and I feel it should be precise and includes the name of the city where this study was conducted (i.e. in the city of Alfenas, Brazil).

2. 1. Keywords need to be placed in alphabetical order and according to MeSH norms.

3. I have checked the similarity index/plagiarism report which shows the similarity to be around 26%. I kindly advise the authors to reduce this similarity index to be below 10%-15% or as per the journal policy (iThenticate report attached).

4. The sample size (child population enrolled) should have been more.

5. More schools would have enrolled in this study.

6. Results seem to be straight-forward, with not many parameters to be checked. Only one table and one figure in the entire study.

7. Discussion: References no. 27, 28 and 29 are missing in the discussion section of the manuscript.

8. Conclusion: When the sample size is small the conclusion would definitely be less significant or non-significant.

9. The second objective of the study was mentioned in the conclusion of the study???

10. 11 authors contributing to such a small study on a smaller child population???

Round 2

Reviewer 1 Report

I am very pleased that the authors responded to the concerns I presented and they have adequately addressed them. 

Reviewer 2 Report

Dear Authors,
Congratulations on your research, I am pleased with the current manuscript.

My best regards.
